# Navigating Virtual Environments Using Leg Poses and Smartphone Sensors

**DOI:** 10.3390/s19020299

**Published:** 2019-01-13

**Authors:** Georgios Tsaramirsis, Seyed M. Buhari, Mohammed Basheri, Milos Stojmenovic

**Affiliations:** 1Information Technology Department, King Abdulaziz University, Jeddah 21589, Saudi Arabia; mesbukary@kau.edu.sa (S.M.B.); mbasheri@kau.edu.sa (M.B.); 2Department of Computer Science and Electrical Engineering, Singidunum University, 11000 Belgrade, Serbia; mstojmenovic@singidunum.ac.rs

**Keywords:** virtual reality, mobile sensors, machine learning, feature selection, movement identification

## Abstract

Realization of navigation in virtual environments remains a challenge as it involves complex operating conditions. Decomposition of such complexity is attainable by fusion of sensors and machine learning techniques. Identifying the right combination of sensory information and the appropriate machine learning technique is a vital ingredient for translating physical actions to virtual movements. The contributions of our work include: (i) Synchronization of actions and movements using suitable multiple sensor units, and (ii) selection of the significant features and an appropriate algorithm to process them. This work proposes an innovative approach that allows users to move in virtual environments by simply moving their legs towards the desired direction. The necessary hardware includes only a smartphone that is strapped to the subjects’ lower leg. Data from the gyroscope, accelerometer and campus sensors of the mobile device are transmitted to a PC where the movement is accurately identified using a combination of machine learning techniques. Once the desired movement is identified, the movement of the virtual avatar in the virtual environment is realized. After pre-processing the sensor data using the box plot outliers approach, it is observed that Artificial Neural Networks provided the highest movement identification accuracy of 84.2% on the training dataset and 84.1% on testing dataset.

## 1. Introduction

Virtual reality (VR) is rapidly expanding, especially after the release of the new generation of VR helmets such as the Oculus Rift and Oculus touch [1]. These devices allow users to experience an alternative reality by utilizing vision, hearing and touch, which offers a high degree of realism and natural interaction with the virtual world. However, one main persistent problem is how to simulate a natural walking experience in a virtual world, in a cost-effective way with minimal hardware requirements.

A recent study [2] showed that there are different locomotion techniques for moving in virtual environments. Among them the most popular are: walking in place, using a controller/joystick, redirected walking, real walking and gesture based moving. Walking using controllers such as keyboards, joysticks and other similar applications have been the traditional ways for moving in virtual worlds (such as non-VR games). While these devices are widely available, they can be difficult to use in VR and are unrealistic. Real walking is the most realistic approach as the user walks naturally in the physical space and the movements are translated to one-to-one movements in the virtual space [3,4]; however, the problem with this approach is that in many cases, the virtual world is much bigger than the available physical world.

Redirected walking is similar to natural walking [5] but each step in the real world corresponds to multiple steps in the virtual world, allowing the users to cover more virtual space in the smaller physical space. If the scaling factor is small, it is not noticeable but if there is a significant mismatch then it becomes unusable. Walking in place is the most popular approach [2] for virtual reality locomotion. The user utilizes machines such as the Virtualizer [6], Stepper Machine [7] and VirtuSphere [8] to walk in virtual environments. While less realistic than real walking, it solves the problem of mapping between the virtual and the real space. The problem is that these machines are cumbersome, unportable, and can be costly.

Moving by making gestures and swinging arms [9] are approaches where pre-defined gestures/movements are recognized and translated to movement. This allow users to perform special actions like flying [7] but it is not realistic for walking since the user is not using their legs, and it requires special hardware such as Microsoft Kinect [10] or armbands and controllers [9]. Other approaches include using a special chair for VR movement [11], moving to the direction that the head is looking at [12] and moving (teleporting) to the virtual location they are pointing at [13].

Our approach is similar to moving by making gestures, but captures leg poses and translates them to movement, employing just standard smart phones for this endeavor, making it a low overhead method of naturally capturing locomotion information in virtual environments.

We aim to provide an efficient, low-cost solution that offers a more natural experience to virtual locomotion than moving by keyboard and mouse while using widely available smartphone sensors instead of dedicated devices. Our solution utilizes the Internal Measurement Unit (IMU) sensors of smartphones placed at the lower parts of user’s legs, for capturing their legs poses. The placement of sensors on lower parts of legs provides more discriminative and reliable data which is easier to classify as seen in [14,15], who employed a similar sensor positioning approach in order to capture the phases of the walking cycle. The data are then analyzed by machine learning approaches which determines the pseudo-movement of the user to move towards a desired direction. Within the context of this paper, we define a pseudo-movement as a limited motion of the user’s leg which translates to a 3D vector representing direction, speed and an acceleration of the virtual avatar. The mapping between the pseudo-movement and the movement in the virtual environment does not have to be linear. The user has to move the legs towards the desired direction and the system will understand the pseudo-movement of the user and simulate the pressing of the corresponding keyboard key. The speed of the avatar movement is depended on the degree to which the user moves the leg in any direction. For example, the more the user moves the leg forward, the faster the avatar will move forward. However, the speed and acceleration are not implemented in the current version. The main difference between this and other approaches is that our approach captures the pseudo-movement and not the actual movement. The proposed approach falls within the family of gesture-based locomotion, employing innovative leg gestures that require minimal physical movement with minimal hardware requirements. This provides a more realistic solution than keyboard/mouse or handheld controller approaches and yet is a more resource efficient solution than virtual reality treadmills. The proposed approach is taking the middle ground to solve the problem of walking in virtual environments.

The structure of the paper is as follows: Section 2 presents the related work. Section 3 provides the theoretical background including the physical description of the system as well as the employed machine learning techniques for locomotion classification. Section 4 describes the methodology used to acquire the data and describes the classification process. In Section 5, the results and comparative analysis for the various techniques applied in this research are provided. Finally, Section 6 discusses some concluding remarks.

## 2. Related Work

Subodha [15] recently presented a method for enhancing the VR experience by mapping a person’s real-time body movements to a virtual world using a combination of Kinect, a full body avatar and Oculus Rift. Despite a few failures in the Kinect skeletal tracker when in sideways poses, it can provide movements mapped with zero latency allowing the user to interact in real time. This however, requires one-to-one mapping between the physical and virtual world which is not always possible. Cakmak et al. [6] presented a method for moving inside virtual environments using a combination of low friction principles and high precision sensors with a special mechanical construction, resulting in a new form of omni-directional treadmill, which they call a “virtualizer” device. Their device uses different sensors to detect the movement of the player. These sensors are located in the ground floor of the virtualizer. The data collected from these sensors are sent to an integrated microcontroller. Using an optimized algorithm, the microcontroller calculates the movements out of the sensor data and sends the movement information to a PC. The virtualizer comes with special software that emulates keyboard actions or controller input. While this approach offers a realistic walking experience and does not need one-to-one mapping between the real and the virtual environments, it requires large expensive hardware. Another approach [16] is moving by using uniforms that contain wearable sensors, such as IMUs, for detecting accurate movements of various leaps. Such approaches are mainly used for creating animations; however, they also require a one-to-one mapping between the virtual and the real world, and using such uniforms or even sets of sensors is expensive and complex. There are other approaches that can allow movement via EMG sensors that capture the movement of muscles and translate this to virtual movement [17], or capture signals from EEG devices that are used to move 3D avatars in virtual worlds [18] but these are complex, inaccurate, expensive, and inefficient approaches that were not designed for virtual reality users and do not provide a realistic feeling of walking to the user.

Apart from simulating movement in virtual environments, approaches that utilize the IMU sensors of mobile phones have been used for calculating Pedestrian Stride Length Estimation [19]. There, the estimated position of pedestrians was based on data from the gyroscope and accelerometer of a six axis IMU placed on the subject’s foot. The IMU was reporting the data to a separate smartphone used mainly for data storage. Their collected data were used to train a Back Propagation artificial neural network (BP-ANN) and then the training system could be used directly without the need for additional training. In [20] similar approaches were used for Step Length Estimation but this time using Handheld Inertial Sensors. The step frequency was calculated using the handheld inertial sensor and analyzed by a Short Time Fourier Transform (STFT). Their work was tested by 10 subjects and showed an error of 2.5–5% over the estimated traveled distance which is comparable with those in the literature, given that no wearable body sensors were used. Both [19,20] tried to solve the problem of calculating the pedestrian covered distance using machine learning approaches and similar sensors that we are proposing in this work. However, this work is totally different not only as it is applied to a different domain but also because unlike [19,20] the user does not have the luxury of walking and significantly changing their physical locations as this would require a one-to-one mapping between the physical and the virtual worlds. While we employ similar sensors and machine learning approaching here, the user only needs to slightly move his leg towards the desired destination while sitting on a chair in order to achieve locomotion using our system, and as a result no physical walking is required. Based on the live input provided, the system needs to make an almost instant decision about the desired direction and simulate the virtual movement by pressing the corresponding keyboard key. This requires a well pre-trained system so that the latency will be minimal and the system can offer an immersive virtual experience to the user.

While still at an early stage, this research points to a new way for solving a major problem in virtual reality. This work resulted in a practical and low cost solution for walking in virtual environments in a realistic way by utilizing the IMU smart phone sensor. Additionally, we have determined which machine learning approach performs best in such a scenario, thereby precisely simulating the pseudo-movement and not the actual movements of users. While this work is primarily intended for movement in virtual reality and virtual environments, the same solution, with minor modifications can be used for controlling other devices such as cars, motorized wheel chairs and so on.

## 3. Theoretical Background

### 3.1. Physical Equipment

A variety of equipment was used for collecting the training data for the system. The most important one is the nine axis IMU sensors that can be found in standard commercial smartphones. The nine axis IMU is a fusion between a three axis gyroscope, a three axis accelerometer and a three axis compass. The accelerometer measures the gravitational acceleration against the *x*, *y* and *z* axes.

The accelerometer measures the acceleration by the utilization of a mass attached to springs and capacitors. When there is a movement the mass will move towards the opposite direction and change the values of the capacitors. By calculating the change of the current of the capacitors, we can estimate the acceleration and the direction. The gyroscope measures angular rate using the Coriolis effect. The sensor works similar to the accelerometer as it is using a mass that will be displaced when a rotation occurs. This displacement will change the values of the capacitors so the rotation can be calculated. Most gyroscopes drift over time and cannot be trusted for a longer timespan but are very precise for a short time. Accelerometers are slightly unstable, but do not drift. The precise angle can be calculated by combining measurements from both the gyroscope and the accelerometer, by using a mathematical approach called a Kalman filter. Magnetometers measure the effect of the earth’s magnetic field on the sensor by using the Hall FA effect or magnetoresistance. In our experiment, we used a Hall effect sensor. In Hall FA sensors, the normal flow of electrons is disturbed by the magnetic field of earth. This will force the electrons to be displaced. The measurement of this displacement can provide us with information about the strength and direction of the magnetic field.

### 3.2. Dataset Description

Using the physical equipment described in Section 3.1, data was collected for both the training and testing phases. During this process, the system was collecting data from nine axes of the IMU for each position of the leg. Furthermore, it was recording the values of the accelerometer, gyroscope and compass for the *x*, *y* and *z* axes. This was performed by 29 subjects and each of them repeated the data collection procedure three times. During the data collection process, the subjects were asked to place their leg in a certain position such as front, back, right and left and record the values of the sensors in these positions. The first column in the dataset, labeled “*d*” is used to capture the direction. Values from 1 to 5 were used to represent front, back, right, middle and left. Columns labeled with “*ax*”, “*ay*” and “*az*” were used to represent to accelerometer values, “*gx*”, “*gy*”, “*gz*” were used to represent the gyroscope values and “*cx*”, “*cy*” and “*cz*” were used to represent the campus values. The dataset collected as above contains 435 observations. With five directions (the resultant positions) considered, there are 87 observations per direction.

### 3.3. Machine Learning Techniques

The initial attempt to solve this problem was implemented based on threshold values of the various leg angles. However, as different users may move their leg to different positions, a static set of rules based only on mobile device angles proved insufficient and machine learning was used instead. Various machine learning techniques have been tested on the dataset obtained from the IMU sensor. The reason for using different machine learning techniques is to find out the suitability of a specific technique for outcome identification. Various factors like accuracy and computation time during execution are considered to make a decision. Computation time is calculated for two situations: training computation time and testing computation time. While, training computation time will not impact virtual environment synchronization, testing computation time is vital to maintain real-time synchronization between the leg movements and the virtual world.

#### 3.3.1. Regression

Regression is used to identify the relationship between dependent and independent variables [21]. In this dataset, we have to find the relationship between the dependent variable “d” and the independent variables “*ax*”, “*ay*”, “*az*”, “*gx*”, “*gy*”, “*gz*”, “*cx*”, “*cy*” and “*cz*”. Regression is represented as:
(1)$$\begin{array}{c} Y=\beta_{0}+\beta_{1}X_{1}+\ldots +\;\beta_{9}X_{9}+\;\epsilon \\ Where\;Y=dependent\;variable\\ X_{i}=independent\;variables,\;i\in \left[1,9\right]\\ \beta_{0}=intercept\;\left(Value\;of\;Y\;when\;X\;is\;zero\right)\\ \beta_{i}=slope\;\left(the\;impact\;of\;Y\;due\;to\;X_{i}\right),i\in \left[1,9\right]\;\\ \epsilon =random\;error \end{array}$$

The coefficient of determination (*r*^2^), which provides the variance in the outcome that is explained by the regression model, is given as:(2)$$\begin{array}{c} r^{2}=\frac{SSR}{SST}\\ Sum\;of\;Squares\;Error\;\left(SSE\right)=\;\sum {\left(Y-\hat{Y}\right)}^{2}\\ Sum\;of\;Squares\;due\;to\;Regression\;\left(SSR\right)=\;\sum {\left(\hat{Y}-\bar{Y}\right)}^{2}\\ SST=SSR+SSE \end{array}$$
where Y is Dependent variable, $\hat{Y}$ is estimated Y, $\bar{Y}$ is mean of Y.

Poisson regression, which is a non-linear regression, considers the dependent variable (*Y*) as an observed count that follows Poisson distribution. The rate of occurrence of an event, *λ*, is based on:(3)λ=exp{Xβ}
where *X* represents the independent variables, and β represents the coefficient of *X*.

So, the Poisson regression equation is: (4)$$P\left(Y_{i}=y_{i}|X_{i},\beta_{i}\right)=\frac{e^{-\exp\left\{X_{i}\beta_{i}\right\}}\exp{\left\{X_{i}\beta_{i}\right\}}^{y_{i}}}{y_{i}!}$$
where $\;\beta_{i},i\in \left[1,9\right]$ represents the nine axis gyroscope input values, captured after the positioning of the leg.

#### 3.3.2. Artificial Neural Networks

Artificial Neural Networks (ANN) is a framework based on the biological neural networks that constitutes the brain. Various algorithms have been applied on this framework to recognize or learn different applicative elements. ANN is formed with input, hidden and output layers, which contain many neurons in each one of them. Input neurons are connected towards the output neurons through some non-linear function. These neurons are connected using edges, which are assigned certain weights. These weights could be increased or decreased based on the learning process. Along with nodes, edges, and weights, a threshold is also used to control the output of the neuron. Basic neural networks are of two types: Supervised neural networks and unsupervised neural networks. In Supervised neural networks, the neural network is trained to produce desired results. In Unsupervised neural networks, neural networks are themselves allowed to make inferences based on input data sets. Initial state is used with appropriate weights $w_{ji}$ to calculate the output. An activation function $\sigma_{j}$ is applied to the output $\xi_{j}^{\left(t\right)}$ to provide an updated output [22].
(5)$$\xi_{j}^{\left(t\right)}=\overset{s}{\underset{i=0}{\sum }}w_{ji}y_{i}^{\left(t\right)}$$
(6)$$y_{j}^{\left(t+1\right)}=\{\begin{array}{cc} \sigma_{j}\left(\xi_{j}^{\left(t\right)}\right), & for\;j\;\in \;\alpha_{t+1}\\ y_{j}^{\left(t\right)}, & for\;j\;\in \;V\backslash \alpha_{t+1} \end{array}$$
where $w_{ji}$ is the weight between nodes, $y_{j}^{\left(t\right)}$ and $\xi_{j}^{\left(t\right)}$ represent output.

#### 3.3.3. K-Nearest Neighbor

As described in [23], Nearest Neighbor uses nonparametric approach where the input vector is classified based on *k* nearest neighbors. Training is required to identify the neighbors based on the distance. Weighted K-Nearest Neighbor (KNN) uses weights based on the distance of the neighbor from the query point *x_q_*. This causes the closer neighbors to have greater weights. But, the processing time to fit for many neighbors can be time consuming. The distance metric applied to two vectors *u* and *v* in Weighted KNN is given as [23]:(7)$$\sqrt{\overset{k}{\underset{i=1}{\sum }}w_{i}{\left(u_{i}-v_{i}\right)}^{2}}$$
where $0<\;w_{i}<1$ and $\sum_{i=1}^{k}w_{i}=1$.

Cubic KNN: The distance metric used in Cubic KNN is cubic distance metric. The cubic distance metric when applied to two vectors *u* and *v* is given as [23]:(8)$$\sqrt[{3}]{\overset{k}{\underset{i=1}{\sum }}{\left|u_{i}-v_{i}\right|}^{3}}$$

Cosine KNN: The distance metric used in Cosine KNN is cosine distance metric. The cubic distance metric when applied to two vectors *u* and *v* is given as [23]:(9)$$1-\frac{u\cdot v}{\left|u\right|\cdot {\left|v\right|}^{\prime }}$$

#### 3.3.4. Decision Trees

Decision Trees are used to classify the given target based on criteria. Decision trees break down the given dataset into smaller subsets, while making the tree with decision nodes and leaf nodes. Decision Trees are built using two entropies, defined as:(10)$$\begin{array}{c} Entropy\;of\;single\;attribute:E\left(S\right)=\;\overset{c}{\underset{i=1}{\sum }}-p_{i}log_{2}p_{i}\\ \mathrm{where}\;p_{i}\;\mathrm{is}\;\mathrm{the}\;\mathrm{probability}\;\mathrm{of}\;\mathrm{variable}\;i,\;\mathrm{which}\;\mathrm{is}\;\mathrm{the}\;\mathrm{set}\;\mathrm{of}\;\mathrm{classes}\;\mathrm{in}\;S\\ Entropy\;of\;two\;attributes:E\left(T,\;X\right)=\;\underset{c\in X,d\in T}{\sum }P\left(c\right)E\left(d\right)\\ \mathrm{where}\;P\left(c\right)\;\mathrm{is}\;\mathrm{the}\;\mathrm{probability},\;E\left(d\right)\;\mathrm{is}\;\mathrm{the}\;\mathrm{entropy},\;T\;\mathrm{is}\;\mathrm{the}\;\mathrm{data}\;\mathrm{set},\;X\;\mathrm{is}\;\mathrm{the}\;\mathrm{attribute} \end{array}$$

The entropy of the target is calculated, and then the dataset could be split using different attributes of the various independent variables. Information gain is calculated by the difference between the entropy before the split and entropy after the split. The one with the largest information gain is used as the decision node of the decision tree. If a branch has zero entropy, then it is a leaf node; else, there is a need for further splitting:(11)Gain(T,X)=Entropy(T)−Entropy(T,X)

#### 3.3.5. Ensemble Decision Trees

It is possible that we might need to combine several imperfect inputs to obtain a better output. The AdaBoost algorithm uses an adaptive boosting approach. In the AdaBoost algorithm, equal weights are assigned initially but the weights of misclassified elements are adjusted and the process is repeated. The final output is based on weights for each classification. AdaBoost initially supports only binary classification. The AdaBoost.M2 algorithm supports multi-class classification. AdaBoost.M2 generates the final hypothesis as [24]:(12)$$\begin{array}{c} h_{fin}\left(x\right)=\arg\underset{y\epsilon Y}{\max}\overset{T}{\underset{t=1}{\sum }}\left(log\frac{1}{\beta_{t}}\right)h_{t}\left(x,y\right)\\ \mathrm{The}\;\mathrm{prediction}\;h_{t}\;\left(\mathrm{at}\;\mathrm{round}\;t\right)\;\mathrm{can}\;\mathrm{have}\;\mathrm{any}\;\mathrm{probability}\;\mathrm{function}\\ \beta_{t}\;\mathrm{is}\;\mathrm{the}\;\text{pseudo-loss}\;\mathrm{in}\;h_{t}\\ x\;\mathrm{represents}\;\mathrm{the}\;\mathrm{independent}\;\mathrm{vairable}\;\left(\mathrm{sensor}\;\mathrm{data}\right)\\ y\;\mathrm{represents}\;\mathrm{the}\;\mathrm{dependent}\;\mathrm{variable}\;\left(\mathrm{position}\;\mathrm{of}\;\mathrm{the}\;\mathrm{leg}\right) \end{array}$$

Bagging uses the approach of combining hypothesis using majority voting. Generally, bagging provides more accurate results but with higher computation time.

### 3.4. Validation Process

Validation of the chosen machine learning approaches is done using accuracy and runtime duration. The computational time of the algorithm training is also calculated, but not taken into account when selecting an approach as it has no impact on gameplay, which utilizes the trained classifier. But, the synchronization between the real world and virtual world movements depends on the computational time of the algorithm testing. Accuracy is calculated as:(13)$$\mathrm{Accuracy}=\;\frac{\left(\mathrm{TP}+\mathrm{TN}\right)}{N}$$
where TP is True Positive count and TN is True Negative count, and N is the number of samples.

## 4. Methodology

The methodology of this research work constitutes of three main stages: (1) Data Collection and Preprocessing stage, (2) Training stage and (3) Testing stage. The general operation is: The acquired data from sensors are preprocessed and then acted upon by different machine learning techniques in the Training Stage. The most efficient machine learning techniques from the Training Stage is selected for the Testing Stage. Figure 1 above shows the system architecture, where both the training and testing procedures are illustrated.

### 4.1. Data Collection and Preprocessing stage

Data is transferred from the IMU sensors to the computer via WiFi. Data obtained from IMU indicate different positions of the subject’s leg. A standard desktop computer performs the locomotion in real time based on the sensor data via a C# based console application. The acquired data is pre-processed for normality. During this phase, outliers which are 1.5 times Inter-Quartile Range away from the first and third Quartile, are removed. The outlier removal is based on the boxplot approach, which removes any value that is far from the 1st and 3rd quartile by 1.5 times the inter-quartile distance. This preprocessed data (Appendix A) is further processed through different machine learning techniques for the purpose of selecting the best fit. This research uses different techniques to classify the given dataset into their respective targeted movements and to measure the accuracy of the system. The movement of the user is predicted with the selected algorithm using data obtained for testing purpose.

The android mobile application that was developed (using Sensor Manager and Sensor classes) was responsible for collecting data from the IMU sensor of the mobile phone and transferring the data via Wi-Fi to a C# application also developed as part of this research project and running on a PC. The C# application has two working modes. One for collecting training data and one for testing. The poses are presented in Figure 2. The sub-figures indicate the poses of the right leg as: (Figure 2A) Front, (Figure 2B) Back, (Figure 2C) Right, (Figure 2D) Middle (Idle), and (Figure 2E) Left.

### 4.2. Training Stage

While training, the application requests the user to place his/her feet in a certain location as instructed by the software and then click the corresponding button. First front, then back, then right, then middle and finally left. The collected data is then evaluated using different approaches. Many salient checks performed on the chosen approaches are:Normality Check: With regards to linear regression, where normality of the data is one of the requirement, is verified using normal Q-Q plot. Further confirmation of normality is handled using Shapiro-Wilk normality test.Salient variables identification: Stepwise regression is used to identify the salient independent variables and see if the adjusted R^2^ value could be improved by just using these, instead of the whole list of independent variables.Handing multiple outcomes: As the number of dependent outcomes are five, binomial based Logistic regression could be not applied. Thus, non-linear regression was applied on the same dataset with a Poisson regression. ANOVA was applied on the results of the Poisson regression to understand their applicability.

Artificial Neural networks with five hidden layers were used since the system has five target outcomes. The target position represented in integer form is converted into five binary bits, each bit representing a single integer value. This conversion is necessary because otherwise the target will be of only one class, leading to a binary decision. Representing multiclass using a single output is different from representing multiclass using multiple binary outputs [25].
Training and Testing Dataset: During the training process, both the training and the validation datasets were used. During the training phase, the weights are adjusted using the training data set and the validation data set is used to control overfitting.Variants of Machine Learning Techniques used: KNN is applied using Weighted KNN, Cubic KNN and Cosine KNN. By varying the number of neighbors, the accuracy of classification is tested. With regards to decision trees, three variants such as Simple (with number of splits equal to 4), Medium (with number of splits equal to 20) and Complex (with number of splits equal to 100) Trees are tested.

### 4.3. Testing Stage

The testing phase was based on using the application with runtime data collected by live user movements transmitted to the PC for classification. The application is then simulated the pressing of the corresponding key. The W key for front, The *S* key for back, the *A* key for left and the *D* key for right. While compass data were available, the application only oriented the user at the beginning and not at runtime. It is assumed that at the start of the application the camera of the virtual environment was facing forward. This is not a problem as VR devices such as oculus can change the orientation of the avatar at runtime based of the direction that the user is looking at. During testing, it was found that users were in need of a few seconds to know how to move their legs to the accepted locations but the ability to move in the virtual world using their legs produced a very immersive experience.

During the testing phase, the subjects were required to wear a smartphone on the lower end of their leg. First the system will auto-calibrate, using the compass data in order to select a front facing direction. All future rotations will be calculated based on the initial front direction. The smartphone communicates with a PC via WIFI and transmits real time IMU data. These data are processed by a trained algorithm which outputs the desired direction.

User satisfaction was measured using a questionnaire, where we asked the 29 students who participated in collecting the training data as well as the 15 users who tested the completed application to shed light about their experience with the system. Table 1 presents the questions and answers.

For the first question, all subjects selected a mark of 5 since they have all completed an optional university level course on 3D games and virtual reality application development, and were very familiar with games and virtual reality. For the second question all subjects replied “keyboard and mouse”. This response was not a surprise as “keyboard and mouse” is the preferred option for first person shooters. For the third question the answers were divided with 22 preferring the proposed approach, 19 handheld controller, three mouse and keyboard. For the fourth question, 26 participants consider the proposed approach more realistic, 17 handheld controllers and one mouse and keyboard. This came as a surprise as we expected the proposed approach to dominate this area. We suspect that the subjects consider handheld controllers as realistic due to the various feedback mechanisms such as force back. In the last question, 31 participants selected 5 while the rest chose 4. The next section evaluates the computational aspects of the proposed approach.

## 5. Results and Discussion

The following methods were tested: Regression, Artificial Neural Networks, K-Nearest Neighbors, Decision Trees, and Ensemble Decision Trees. The data without any pre-processing gave results as shown in Table 2. The results in Table 2 are used to show the importance of preprocessing. Else, the accuracy is degraded.

Accuracy of the dataset is considered for two cases: with and without outlier removal. The pre-processing approach of outlier removal removes 33% of the dataset; thus reducing the observations to 290. Among the five directions, back position has most number of outliers, and thus only 56% of this observation is present in the post-processed dataset. In more detail, 56 rows for “front”, 49 for “back”, 52 for “right”, 69 for “middle” and 64 for “left”.

### 5.1. Regression

#### 5.1.1. Linear Regression

Applying linear regression, gave the following outcome:(14)$$\begin{array}{cc} d=1.352116+ & 0.180844\;\times ax+0.207074\;\times ay-0.058567\;\times az+0.443486\;\times gx\\ & -0.021408\;\times gy+0.004335\;\times gz-0.002796\;\times cx-0.004092\;\times cz \end{array}$$

The adjusted R^2^ value obtained for the linear regression is 0.2588, which is low and thus the applicability of linear regression for this dataset is questionable. Figure 3 shows the normal Q-Q plot.

From the normal Q-Q plot, the normality assumption required for linear regression is not satisfied. Using Shapiro-Wilk normality test, the *p*-value is 3.856 × 10^10^, which is lower than the α value, indicates that the null hypothesis (that the sample comes from a population that is normally distributed) is rejected. The same was confirmed using the Kolmogorov-Smirnov test, where the obtained *p*-value was 2.995 × 10^5^. Thus, the non-applicability of linear regression to this dataset is confirmed.

Stepwise regression indicated that *ax*, *ay* and *az* are the only salient independent variables. Using these two variables, linear regression returned an adjusted R^2^ value of 0.2661, confirming the non-applicability of linear regression to this dataset:(15)d=1.45693+0.18771 ×ax+0.19714 ×ay−0.06890 ×az

#### 5.1.2. Non-Linear Regression

Poisson regression could handle multiclass dependent outcomes, and is represented as: the dependent variable “*d*”, which follows a Poisson distribution with rate
(16)$$\begin{array}{cc} \lambda =\exp\{0.5649771 & +0.0613476\times ax+0.0636902\times ay-0.0186062\;\times az\\ & +0.1644936\;\times gx+\;0.0171299\;\times gy+0.0174577\;\times gz\\ & -0.0008213\;\times cx-0.0015632\;\times cz\} \end{array}$$

Residual deviance of 110.06 was obtained with 204 degrees of freedom. The *p*-value obtained was 1. The value 110.06/204 is smaller than 1, so the model seems to fit the dataset. The residual plot, as shown in Figure 4, indicates that the chosen regression is suitable, but there are some outliers which needs to be addressed.

The obtained Pseudo R^2^ value for ANOVA was 0.2626259. The obtained *p*-value using the Pearson method was 1.142181 × 10^10^. This indicates a lack of fit for this regression. Thus, we move on to apply other machine learning techniques.

### 5.2. Artificial Neural Networks

Training of the neural network was achieved with 20 iterations. The best validation result was attained at the 14th epoch, shown in Figure 5.

The confusion matrix provided in Figure 6 gives details about how each class of this multiclass dataset is learnt or identified by the neural network system. The classes here represent different directions (front (1), back (2), right (3), middle (4) and left (5)).

Figure 6 indicates the training and testing results. Overall training results indicate 84.2% accurate classification, but the misclassification of classes is higher for target class two, in the training dataset. The testing results were also very close with 84.1%, using 15.9% of the dataset of testing.

### 5.3. K-Nearest Neighbors

We have used 14 as the number of neighbors since reducing this number further decreases accuracy. For the Weighted KNN algorithm, increasing the number of neighbors to 19 increases the accuracy but larger K values than this reduce accuracy. We have used normalized data, since the accuracy decreases without normalization. Overall, the Weighted KNN has the highest accuracy as 78.6%.

Comparing the accuracies mentioned in Table 3, weighted KNN outperformed other approaches. Weighted KNN achieved an overall accuracy of 78.6% when the number of neighbors is 19. Cubic KNN achieved overall accuracy of 73.4% when the number of neighbors is 15. Cosine KNN achieved overall accuracy of 76.2% when the number of neighbors is 17. The respective confusion matrices for different KNN approaches is shown in Figure 7 and Figure 8.

### 5.4. Decision Trees

Medium Trees resulted in better accuracy than other two, with 71.7%, while Simple Tree had an accuracy of 62.4% and that of Complex Trees was 71.4%.

The confusion matrices obtained for different decision tree approaches are shown in Figure 9 and Figure 10. Figure 11 shows the decision tree obtained from the Medium Tree, which gave quite similar accuracy to that of the Complex Tree.

### 5.5. Ensemble Decision Trees

AdaBoost and Bag approaches were used. AdaBoost achieved accuracy of 75.9%. Bag achieved an accuracy of 80.3%. AdaBoost took 2.250457 seconds while Bagging took 2.159735 seconds to train the classifier. This time taken is only for offline training. Figure 12 shows the confusion matrix for the two Ensemble approaches used.

### 5.6. Comparison of Different Approaches

Table 4 and Table 5 summarize the results obtained from various machine learning techniques. Selection of any machine learning algorithm depends on the trade-off between the computation time and accuracy. Based on the results shown in Table 5, the ANN achieves the highest accuracy with reasonable computation time. The next achieved accuracy is with Ensemble Bagged Trees but with higher computation time. Depending upon the pace of the virtual game, a much faster approach could be considered with reduction in accuracy.

The ANN can estimate the desired direction with 84.1% accuracy, within 0.0192 milliseconds. The speed of the calculation is important as any introduction of lags (respond delay) will have a very negative impact to the virtual experience. As currently the accuracy of the classifier is not 100%, the system may misinterpret some legs poses, however after a short period of time the user gets used to the valid physical locations/poses and tend to place his/her leg at the right position.

### 5.7. Limitations and Future Work

It is always possible that a user can move towards front-right or front-left and so on. In this scenario, we wish to see whether our system can identify the diagonal movements or not. Based on the preliminary testing results obtained using diagonal movements, regression (both linear and non-linear), artificial neural networks and KNN were tested. Regression which was poor in the original dataset (Appendix A), also could not identify the diagonal movements properly. Artificial neural networks gave outputs by classifying front-right as front and back-left as back, but others were generally misclassified. KNN classified the front-left as front and so on, as there were only five classes in the initial dataset. Overall, the system learns appropriately for the four directions (front, back, right and left) dataset but with diagonal movements (front-left, front-right, etc.), only artificial neural network was able to identify certain diagonal movements. These diagonal movements are to be studied as future work.

The main limitation of this work is that the data were collected for the forward, backward, right, center and left and not for the diagonal movements, jump and crouch. Jumping and crouching cannot be detected by the current version of the system, but the diagonal movements can be identified by the neural network and it returns two outputs. Jumping could be implemented by moving the leg in an upward position for the duration of a jump while crouching can be implemented by bending the leg forward. The speed could be calculated by moving the leg to the extreme ends of the corresponding movement.

Another limitation is that the direction is calculated after the placement of the leg to a position and not during the movement. This forced the identification process to start after the user’s physical movement causing a small delay. In order to provide an even more immersive experience, data can be collected during the movement and not only at the end. This can allow prediction of the desired movement and eliminate the delay, offering an even better experience. This experience can be further enhanced by adding walking clues such as audio messages, as proposed in [26].

## 6. Conclusions

This paper presents a new way for walking in a virtual environment. Unlike other approaches, this work allows the user to move in the virtual world simply by moving their leg towards the desired direction. This can happen while the users sit on a chair. The system interprets your physical movement and converts it to virtual movement by identifying the desired pseudo movement. To prove that our proposal is possible, we develop two applications that we primarily used for collecting training data and we analyzed these data with a number of machine learning techniques.

Based on the abovementioned results, it could be concluded that Artificial Neural Networks and Ensemble Bagged approach performed better than other considered approaches. The usage of regression for this dataset is questionable. Both ensemble bagged trees and ANN can be used in future development of similar applications.

## Figures and Tables

**Figure 1 sensors-19-00299-f001:**
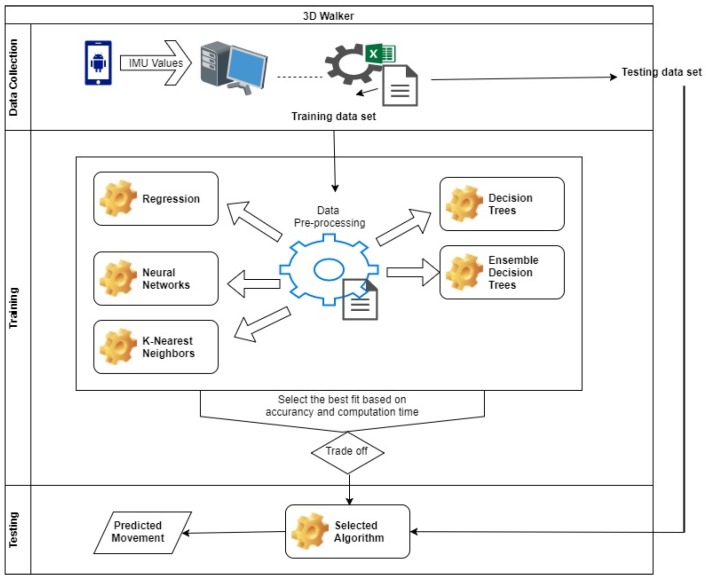
System Architecture.

**Figure 2 sensors-19-00299-f002:**
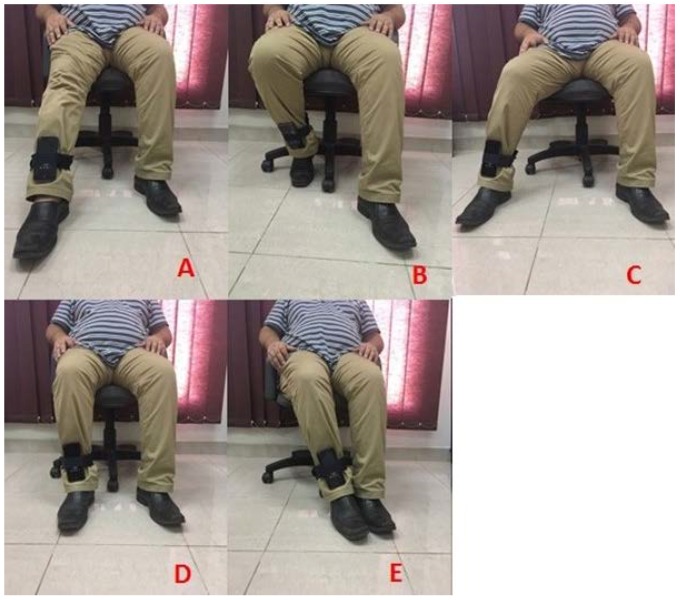
Data Collection Process. (**A**) Front; (**B**) Back; (**C**) Right; (**D**) Middle (Idle); and (**E**) Left.

**Figure 3 sensors-19-00299-f003:**
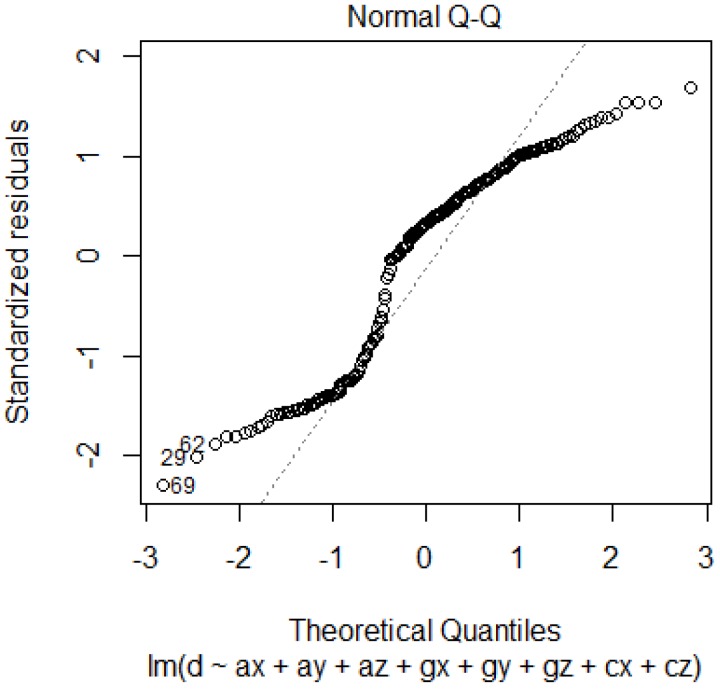
Normal Q-Q Plot.

**Figure 4 sensors-19-00299-f004:**
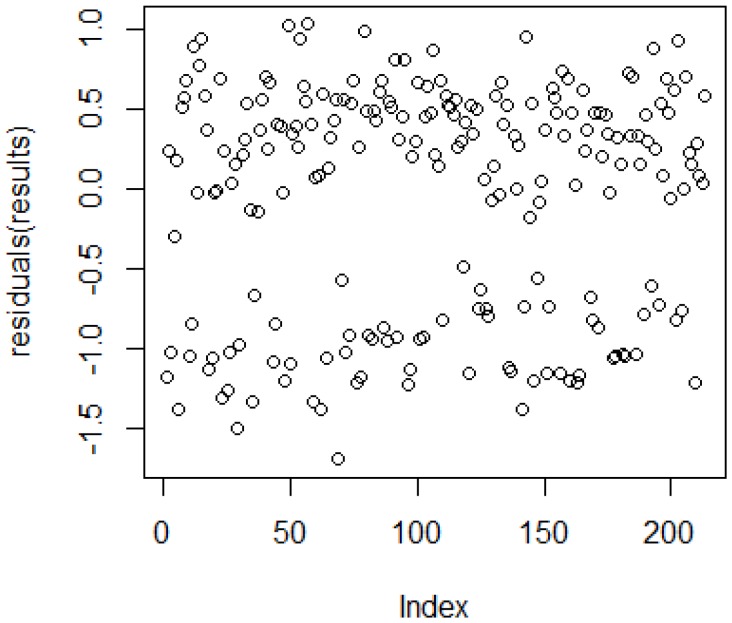
Residual Plot.

**Figure 5 sensors-19-00299-f005:**
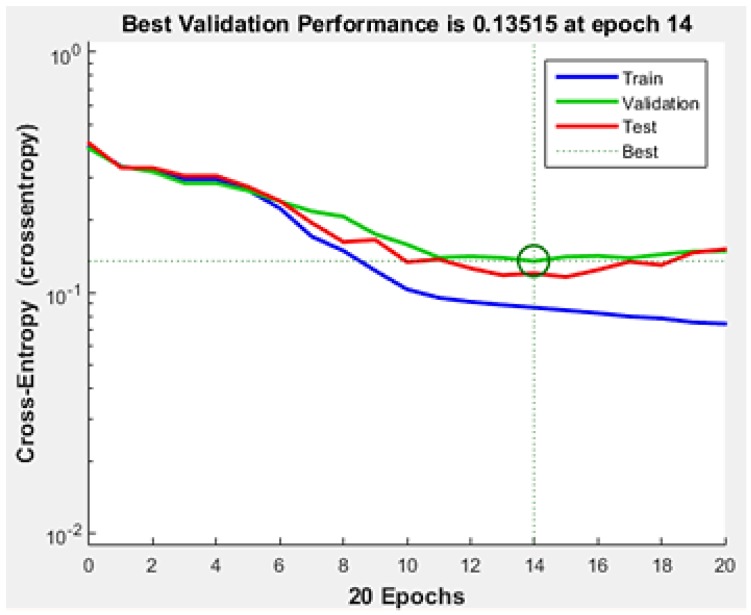
Validation Performance.

**Figure 6 sensors-19-00299-f006:**
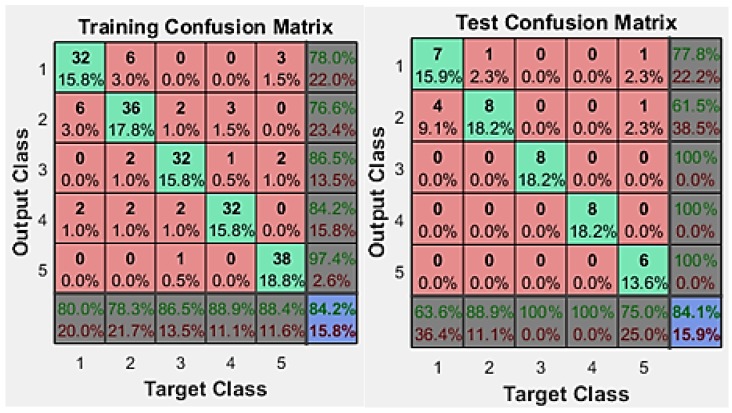
Confusion Matrix for Training Dataset.

**Figure 7 sensors-19-00299-f007:**
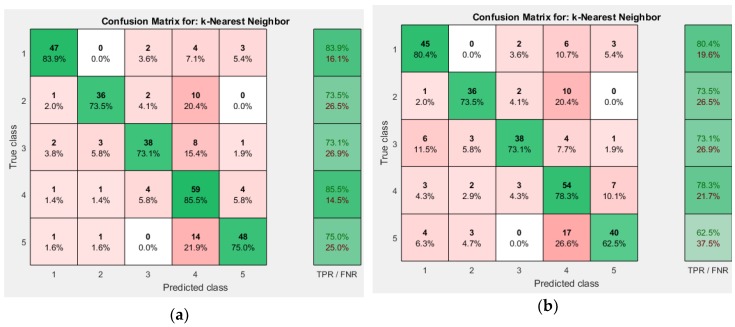
Confusion Matrix (**a**) Weighted KNN; (**b**) Cubic KNN.

**Figure 8 sensors-19-00299-f008:**
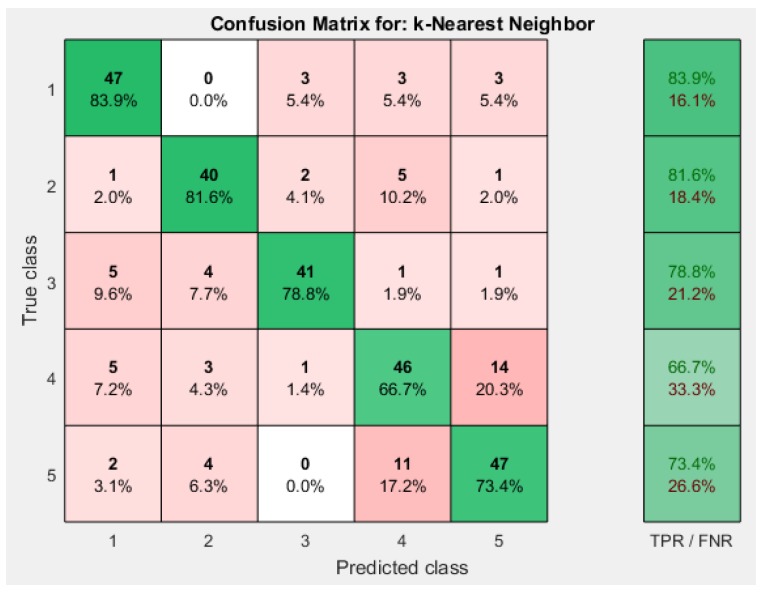
Cosine KNN Confusion Matrix.

**Figure 9 sensors-19-00299-f009:**
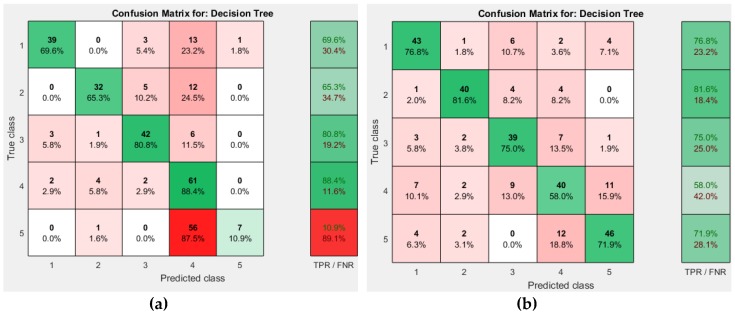
Decision Tree Confusion Matrix. (**a**) Simple Tree; (**b**) Medium Tree.

**Figure 10 sensors-19-00299-f010:**
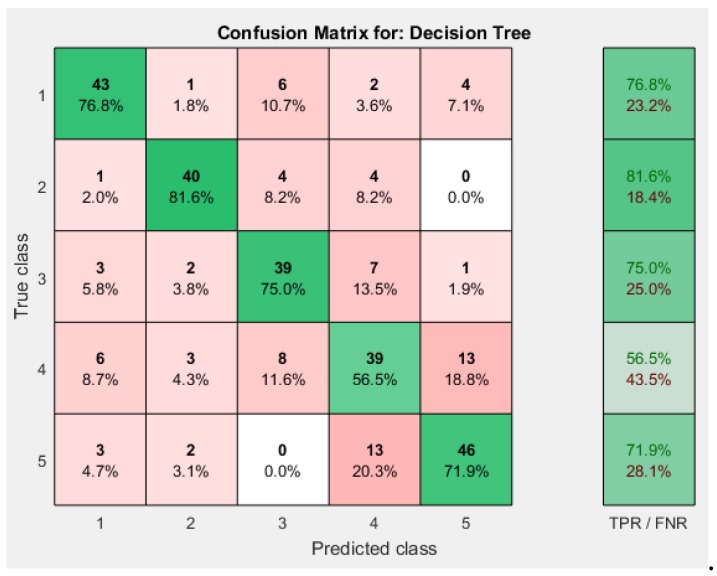
Decision Tree (Complex Tree) Confusion Matrix.

**Figure 11 sensors-19-00299-f011:**
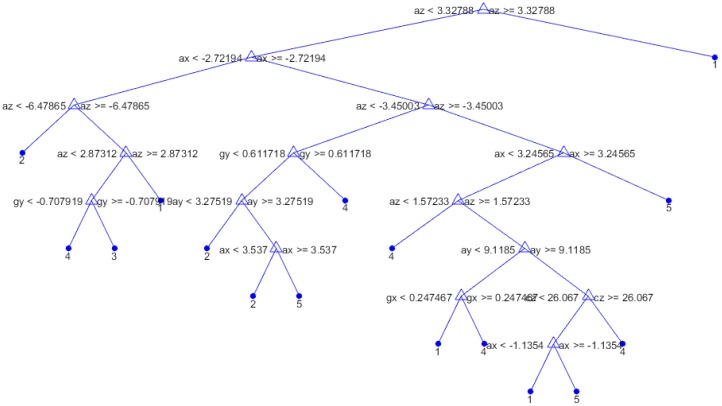
Decision Tree for Medium Tree.

**Figure 12 sensors-19-00299-f012:**
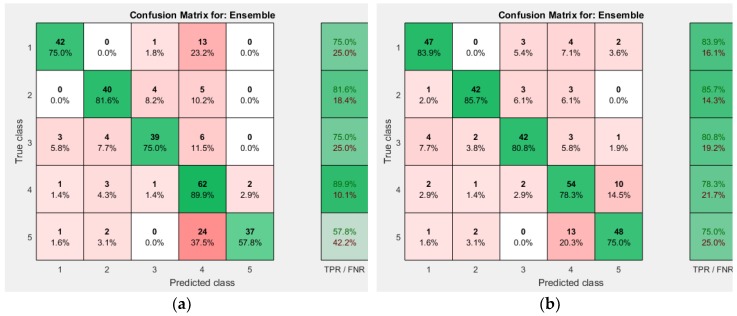
Ensemble Decision Tree Confusion Matrix. (**a**) AdaBoost; (**b**) Bag.

**Table 1 sensors-19-00299-t001:** Evaluation of User Satisfaction.

Question	Score
(1) Are you interested in first person computer games and virtual reality applications? Please grade from 1 to 5. 5 is very, while 1 is not at all.	Average = 5.0
(2) What is your preferred game controller for first person games)? (a) Keyboard and mouse (b) Gamepads (c) The proposed approach (d) Other.	(a) 44/44 (b) 0/44 (c) 0/44 (d) 0/44
(3) What is your preferred controller for virtual reality applications while sitting on a chair? (a) Keyboard mouse (b) Hand held controllers (c) The proposed approach (d) Other.	(a) 3/44 (b) 19/44 (c) 22/44 (d) 0/44
(4) Which of the above gives you a more realistic experience in VR applications while sitting on a chair? (a) Keyboard and mouse (b) Hand held controllers (c) The proposed approach (d) Other.	(a) 1/44 (b) 17/44 (c) 26/44 (d) 0/44
(5) How easy was it to move in virtual words using the proposed approach? Please grade from 1 to 5. 5 is very, while 1 is not at all.	Average = 4.4

**Table 2 sensors-19-00299-t002:** Accuracy without data pre-processing.

	Regression	ANN	KNN	Decision Trees	Ensemble
Accuracy	Lack of fit	67.7%	70.3%	69.4%	72.2%

**Table 3 sensors-19-00299-t003:** K-Nearest Neighbors Comparison.

	14	15	16	17	18	19	20	21
Weighted KNN	77.9%	77.6%	78.3%	77.9%	76.9%	**78.6%**	76.2%	77.6%
Cubic KNN	**73.4%**	73.4%	71.0%	71.0%	72.1%	69.7%	69.0%	70.0%
Cosine KNN	75.5%	74.8%	75.2%	**76.2%**	74.8%	74.8%	75.2%	75.9%

**Table 4 sensors-19-00299-t004:** Results Comparison (Training data).

	Artificial Neural Networks	KNN	Decision Trees	Ensemble Decision Trees (Bag)
Accuracy	84.2%	78.6%	71.7%	80.3%
Training Time (Sec)	0.0313	0.036150	0.024026	2.159735

**Table 5 sensors-19-00299-t005:** Results Comparison (Testing data).

	Artificial Neural Networks	KNN	Decision Trees	Ensemble Decision Trees (Bag)
Accuracy	84.1%	76.9%	72.76%	74.83%
Runtime execution (1 classification) (Sec)	0.0192	0.002264	0.000908	0.246718

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
