# Peer review of "Navigating Virtual Environments Using Leg Poses and Smartphone Sensors"

_sensors, 2019, doi:10.3390/s19020299_

Round 1
Reviewer 1 Report
The paper describes an evaluation of machine learning techniques for leg pose detection from smartphone sensors attached to the user's leg. The leg pose is used as an interaction technique for locomotion in virtual environments.
Although it is an interesting interaction technique, the paper suffers from a series of issues that make it hard to understand the purpose and contribution of the work.
Although the authors mention various kinds of locomotion techniques for VR (real walking, walk-in-place, controllers, etc.), it is not clear how they position their own technique. In the introduction, the authors state that their novel approach allows users to "actually moving their legs". (This in itself is not novel). In addition, authors state that their solution "offers a more natural experience". (More natural than what?) This seems to point to a walk-in-place kind of technique, but from the rest of the paper we learn that leg movements are not natural (they don't attempt to mimic real walking) and that legs are used as a kind of joystick!
It would be important to know where the authors position their interaction technique in terms of expected walking realism since they explicitly mention that offering a more natural experience to VR is a property of their solution.
The paper seems to ignore the various kinds of walk-in-place techniques that have been developed. Although walk-in-place is mentioned, the paper seems to indicate that the proposed solution is a novel approach sitting in between using handheld controllers and treadmills.
In addition, it would be important to know the context in which the proposed solution is to be used. Although not clearly discussed, the proposed solution only works when users are sitting down (it would be hard to use the technique during long periods of time). This limits the applicability of the technique (although that in itself is not a bad thing). What applications would benefit from this approach?
The paper also lacks definitions for (apparently important) concepts: what is a pseudo-movement?
Regarding the machine-learning techniques, is it not clear why the compass values would be of use for the kind of leg poses to be detected. It is also not clear why a machine-learning technique is required in this situation. The technique seems to detect only static leg poses, which correspond to different tilt angles of the mobile device. Why can't the leg pose be estimated from accelerometer values only?
One important aspect of gesture detection for locomotion is the recognition delay: after the user decides to move, how long will it take the system to recognize that intention? (i.e., how long after the users starts moving his leg.)
In summary, I think the paper reports interesting work, but requires a better contextualization of the problem and a discussion of limitations of the technique when compared to other similar alternatives.
Author Response
Reviewer Remarks | Author feedbacks |
In the introduction, the authors state that their novel approach allows users to "actually moving their legs". (This in itself is not novel)
Authors state that their solution "offers a more natural experience". (More natural than what?) The paper seems to ignore the various kinds of walk-in-place techniques that have been developed. Although walk-in-place is mentioned, the paper seems to indicate that the proposed solution is a novel approach sitting in between using handheld controllers and treadmills. | We believe that this misconception is solved after reordering certain parts of the introduction, as mentioned by Reviewer 3. |
This seems to point to a walk-in-place kind of technique, but from the rest of the paper we learn that leg movements are not natural (they don't attempt to mimic real walking) and that legs are used as a kind of joystick! | The usage of leg like a joystick make the gamer with free hands so that he/she can use his/her hands to do certain actions in the game instead of just moving the character. |
Where the authors position their interaction technique in terms of expected walking realism since they explicitly mention that offering a more natural experience to VR is a property of their solution?
| It is claimed that it is more realistic than using mouse and keyboard, as usually, humans move using their legs and not their hands. This can also have other potential benefits for applications such as games since it allows the games to user their hands for other actions, while sitting on a chair.
There are more realistic aproach than the one proposed in this paper, but they require dedicated devices and not a commonly device like mobile phones. Another advantage with the proposed approach is that the user does not have to move his/her leg all the time (like with natural movement) as our approach is similar to driving a car. This enables our approach to be used for longer times without causing fatigue to the user.
We believe that our approach is taking the middle ground.
The text in the introduction has been reorganized to better communicate these concepts. |
Proposed solution only works when users are sitting down (it would be hard to use the technique during long periods of time). This limits the applicability of the technique (although that in itself is not a bad thing). What applications would benefit from this approach? | All gamers play games while sitting in the chair. So, the applicability of this approach is justified. This approach will also release their hands of the gamer and thus they could use their hands for other actions within the game. |
The paper also lacks definitions for (apparently important) concepts: what is a pseudo-movement? | Pseudo-movement in this paper indicates fake movement, which is considered equivalent to real physical movement in this paper. |
The technique seems to detect only static leg poses, which correspond to different tilt angles of the mobile device. Why can't the leg pose be estimated from accelerometer values only? Use of Compass? | Direction is indicated using Compass and the starting position is calibrated using Compass. Accelerometer and gyroscope are used in a combination to reduce noise and increase efficiency. |
After the user decides to move, how long will it take the system to recognize that intention? (i.e., how long after the users starts moving his leg.) | This is shown in Table 3. |
Reviewer 2 Report
The paper is well written and interesting. However, there are some issues with the experiment description that have to be addressed prior to its publication.
The authors say that four directions can be recognized. However, when the smart phone is attached to the right leg, the user can only move to left direction when he is sitting, as shown in Figure 2. How do the authors cope with this problem when the user is standing up? I guess this kind of movement is more difficult to recognize than the others. It would be interesting to show the performance of the classifiers for each of the four directions. Also, it is not clear until the experiment description if the authors are trying to recognize poses or movements. This has to be made clear much earlier.
I'd not consider K-Nearest Neighbor as a supervised learning technique. So, how do the authors train them? How do they get clusters aligned with the four directions? This should by explained in more depth.
Why are the Artificial Neural Networks not mentioned in section 3.3?
In page 10, line 324, the authors say that the target position represented in integer form is converted into five binary bits. This should also be explained in more depth, since the ANN deals with the same number of labels in both cases (four directions).
How are the results of the experiments validated? Do the authors employ cross-validation? I guess the authors employ two different training and testing subsets. How are the samples selected for each of them?
I think there are other cheaper devices that can provide the same information. Even the VR controllers themselves could be used for the same exact purpose. What are the benefits of employing a smart phone?
Some minor issues:
Page 4, line 166: What are the campus values?
Page 6, line 236: Please revise the sentence "During this phase, outliers..."
Author Response
Reviewer 2 | |
The authors say that four directions can be recognized. However, when the smart phone is attached to the right leg, the user can only move to left direction when he is sitting, as shown in Figure 2. How do the authors cope with this problem when the user is standing up? I guess this kind of movement is more difficult to recognize than the others. It would be interesting to show the performance of the classifiers for each of the four directions. Also, it is not clear until the experiment description if the authors are trying to recognize poses or movements. This has to be made clear much earlier.
However, when the smart phone is attached to the right leg, the user can only move to left direction when he is sitting, as shown in Figure 2. | The right leg could be moved to any direction. The movement of the right leg towards left direction is shown in Figure 2(E).
|
How do the authors cope with this problem when the user is standing up? I guess this kind of movement is more difficult to recognize than the others. | As also indicated in our response to Reviewer 1, it can be used for gaming as well as other approaches where sitting possible is dominant. |
It would be interesting to show the performance of the classifiers for each of the four directions. | As an example, confusion matrix as shown in Figure 7 and 8, shows how each position is recognized by the machine learning approaches used. |
Also, it is not clear until the experiment description if the authors are trying to recognize poses or movements. This has to be made clear much earlier.
| The introduction has been re-organized and it is now more clear. |
I'd not consider K-Nearest Neighbor as a supervised learning technique. So, how do the authors train them? How do they get clusters aligned with the four directions? This should by explained in more depth. | Added in Section 3.3.3. |
Why are the Artificial Neural Networks not mentioned in section 3.3? | Added in Section 3.3.2. |
In page 10, line 324, the authors say that the target position represented in integer form is converted into five binary bits. This should also be explained in more depth, since the ANN deals with the same number of labels in both cases (four directions). | Added in Methodology Section |
How are the results of the experiments validated? Do the authors employ cross-validation? I guess the authors employ two different training and testing subsets. How are the samples selected for each of them? | Sample selection is done as: Training (network is adjusted according to the error), validation (halt training when generalization stops improving) and testing (independent measure of network performance during and after training) set. |
I think there are other cheaper devices that can provide the same information. Even the VR controllers themselves could be used for the same exact purpose. What are the benefits of employing a smart phone? | Smartphones are commonly found as compared to VR controllers. Thus, the approach followed is viable. |
Reviewer 3 Report
Navigating virtual environments using leg poses and smartphone sensors
Page 1, line 25: There could be included more keywords, such as "feature selection", "movement identification", among others.
Page 1, line 29: "virtual reality" -> VR
Page 1, line 29: "Oculus Rift and Oculus touch" -> References?
Page 1, line 34: "We aim to provide an efficient, low-cost solution that offers a more natural experience to virtual locomotion. Our solution utilizes" -> Before explaining the solution proposed in the presented work, it is necessary to present and develop the problem. This paragraph should be relocated.
Page 1, line 43: "This provides a more realistic solution than 43 keyboard/mouse or handheld controllers approaches and yet more resource efficient solution that virtual reality treadmills. " -> Reference?
Page 3, line 114: "which is considered good" -> in comparison with which other error value?
Page 3, line 138: "Internal Measurement Unit (IMU)" -> It is not necessary to specify a term more than once.
Page 3, Section Physical Equipment: Which specific mobile equipment was used?
Page 4, line 175: "to identify the relationship"
Page 4, line 175: Reference?
Page 4, line 178: "Linear regression is used to find linear relationship between the dependent and independent variables." -> This line is repetitive.
Page 4, line 180: "? = ?0 + ?1?1 + ⋯ + ?9?9 + ? " -> It could be helpful to list the equations in order to facilitate their referencing.
Page 4, line 180: "X = independent variable" -> There is not a X variable in the equation.
Page 4, line 180: What does β9 mean?
Page 4, Section Machine Learning Techniques: Artificial Neural Networks are not described among the techniques presented in this section, which is very important taking into account that is the technique that presented the best results.
Page 5, line 183, 184, 185, 186, 191: List the equations.
Page 5, line 184, 185: What do " " and " " mean?
Page 5, line 193: "KNN" -> K-Nearest Neighbor (KNN)
Page 5, line 194: "the processing time is slightly longer." -> In comparison with what?
Page 5, line 195: "u", "v" -> variables should be written in cursive letter to differentiate them from the rest of the text.
Page 5, line 200: List the equation.
Page 5, line 199: "u", "v" -> Change to cursive letter
Page 5, line 203: List the equation.
Page 5, line 205: Reference?
Page 5, line 208: List the equation. What do "c" and "i" mean?
Page 5, line 209: List the equation. What do "X" and "T" mean?
Page 6, line 215: List the equation.
Page 6, line 218: "AdaBoost algorithm uses an adaptive boosting approach." -> Reference?
Page 6, line 223: List the equation. What does "T" mean?
Page 6, line 228: "The computational time of the algorithm training is also calculated, but not taken into account..." - > What was the purpose of calculating the computational time?
Page 6, line 230: List the equation.
Page 6, Section Methodology: This section needs to be rewritten, it is very difficult to follow the thread of the text, it is not organized and it does not describe all the steps that were followed, according to the methodology described in the previous sections.
Page 6, line 236: "The acquired data is pre-processed for normality" -> What this line refers to? Which were the steps for the data preprocessing?
Page 6, line 237: "outliers which are 1.5 times Inter-Quartile Range away from the first and third Quartile..." -> What happen with those outliers?
Page 6, line 237: "The data is processed through different machine learning techniques for the purpose of selecting the best fit." -> This line is repetitive.
Page 6, line 238: "An Artificial Neural Network was selected as the most suitable machine learning technique due to its highest accuracy, 84.1% (on testing data) and 84.2% on training data. " -> This line corresponds to the conclusions section.
Page 7, Figure 1: This Figure needs to be better described, specifying in more detail each block of the architecture.
Page 7, line 244: "The android mobile application that was developed..." -> How was this appplication developed? It is necessary to explain the details of the application.
Page 7, Figure 2: The foot figure does not describe what is presented in the image, besides it needs to be located just below the figure. It is necessary to better describe this figure, specifying each of its parts.
Page 8, line 256: "the S key..."
Page 8, line 259: "virtual reality" -> VR
Page 8, Section Results: This section needs to be rewritten, it is very difficult to follow the thread of the text, it is not organized and it does not describe the results obtained. It contains a lot of information that corresponds to another sections, such as the methodology, discussion and conclusions sections.
Page 8, line 265: "Data obtained from IMU indicate different positions of the subject’s leg. This research uses different techniques to classify the given dataset into their respective targeted movements and to ..." -> This text corresponds to the Methodology section.
Page 8, line 268: "The data without any pre-processing gave results as shown in Table 1." -> It is necessary to mention in the Methodology section that the testing of the data is performed using preprocessed and not preprocessed data. On the other hand, what do these results represent? They are results of what?
Page 8, line 272: "As the accuracy did not meet our expectations..." -> Which were those expectations?
Page 8, line 273: "The outlier removal is based on the boxplot approach, which removes any value that is far from the 1st and 3rd quartile by 1.5 times the inter-quartile distance. This pre processing approach ..." -> This paragraph corresponds to the Methodology section.
Page 8, line 281: "In order to find the relationship between the independent variables (like ax, ay, az, gx, gy, gz, cx 281 and cz) and the dependent variable (d), regression was used. " -> This line is repetitive.
Page 8, line 288: "The adjusted R2 value" -> It should be "R^2"
Page 8, line 288: "which is low and thus the applicability of linear regression for this dataset is questionable." -> This line corresponds to the Discussion section.
Page 8, line 289: "To verify this outcome, a normal Q- Q plot was tested." -> This line corresponds to the Methodology section.
Page 9, line 293: "From the normal Q-Q plot, the normality assumption required for linear regression is not satisfied." -> This line corresponds to the Discussion section.
Page 9, line 289: "To confirm the normality test, a Shapiro-Wilk normality test was performed. The p-value of 294 3.856e-10, which is lower than the alpha value, indicates that the null hypothesis (that the sample comes from a population that is normally distributed) is rejected..." -> This paragraph corresponds to the Methodology section.
Page 9, line 299: "Further, stepwise regression was used to identify the salient independent variables and see if the adjusted R2 value could..." -> This paragraph corresponds to the Methodology section. (It should be "R^2").
Page 9, line 302: "adjusted R2 value" -> It should be "R^2"
the Methodology section. (It should be "R^2").
Page 9, line 302: "d=1.45693+0.18771 ×ax+0.19714 ×ay−0.06890 ×??" -> What does it represent? It is not described in the text.
Page 9, line 305: It is necessary to mention in the last sections that a non-linear technique is also used, as well as the description of the technique.
Page 9, line 306: "As the number of dependent outcomes are five, binomial based Logistic regression could be not applied. Thus, non-linear regression ..." -> This paragraph corresponds to the Methodology section.
Page 9, line 310: What does this equation represent?
Page 9, line 312: "204 degrees of freedom." -> It is necessary to explain why was chosen this number of degrees of freedom.
Page 9, line 299: "so the model seems to fit the dataset. The residual plot 313 indicates that the chosen regression is suitable, but there are some outliers which needs to be addressed..." -> This paragraph corresponds to the Discussion section.
Page 10, line 318: "ANOVA was applied on the results of the Poisson regression to understand their applicability." -> This line corresponds to the Methodology section.
Page 10, line 319: "The obtained Pseudo R2 value" -> It should be "R^2".
Page 10, line 320: "This indicates a lack of fit for this regression. Thus, we move on to apply other machine learning techniques." -> This line corresponds to the Discussion section.
Page 10, line 323: "Artificial Neural networks with five hidden layers were used since the system has 5 target outcomes. The target position represented in integer form is converted into five binary bits, each bit representing a single integer value..." -> This paragraph corresponds to the Methodology section.
Page 10, line 327: "During the training process, both the training and the validation datasets were used. During the training phase ..." -> This paragraph corresponds to the Methodology section.
Page 10, line 331: What does Figure 5 represent? It it not described in the text. The foot figure does not describe the image.
Page 11, line 333: Which are does classes? How are they represented in the confusion matrix?
Page 11, line 337: "Overall training results indicate 84.2% 337 accurate classification..."-> Where is this results indicated?
Page 11, line 337: "The testing results were also very close with 84.1%, using 15.9% of the dataset of testing."-> Where are those results indicated?
Page 11, line 341: "K-Nearest neighbors" -> KNN
Page 11, line 341: "K-Nearest neighbors was applied using Weighted KNN, Cubic KNN and Cosine KNN. By varying the number of neighbors, ..." -> This paragraph corresponds to the Methodology section.
Page 11, line 347: Table 1 is not cited in the text.
Page 11, line 349: "Comparing the accuracies, weighted KNN outperformed other approaches. Weighted KNN achieved an overall accuracy of 74.6% when ..." -> This paragraph corresponds to the Discussion section.
Page 12, line 354: Figure 7 and Figure 8 are not mentioned nor described in the text.
Page 12, line 358: "Three variants of decision trees, such as Simple (with number of splits equal to 4), Medium (with 358 number of splits equal to 20) and Complex (with number of splits equal to 100) Trees were tested." -> This line corresponds to the Methodology section.
Page 12, line 360: "Medium Trees resulted in better accuracy than other two, with 71.7%, while Simple Tree had an accuracy of 62.4% and that of Complex Trees was 71.4%." -> This line corresponds to the Discussion section.
Page 12, line 362: Figure 9 and Figure 10 are not mentioned nor described in the text.
Page 13, line 366: "which gave quite similar accuracy to that of the Complex Tree." -> This line corresponds to the Discussion section.
Page 13, line 367: It is difficult to see the text that is inside Figure 11.
Page 13, line 372: "AdaBoost took 2.250457 seconds while Bagging took 2.159735 seconds to train the classifier." -> If the processing time of an algorithm is mentioned, it is necessary to mention the one of all the used ones; however, if the processing time did not fulfill any function for the purpose of this work, it is not necessary to mention it.
Page 14, line 374: Figure 12 is not mentioned nor described in the text.
Page 14, Section Discussion: This section needs to be rewritten. The discussion of the results is not presented in an adequate way, besides the section is incomplete. It is important to restructure it, eliminate the text that does not correspond and keep an order that is consistent with the results.
Page 14, line 378: "Based on the results shown"
Page 14, line 379: "with reasonable computation time." -> Which is that time? Why is it necessary to mention this since the computation time was not taking into account for the selection of an algorithm?
Page 14, line 382 and 384: Table 2 and Table 3 belong to the Results section, besides, these tables do not present the correspondent results of the regression technique.
Page 14, line 390: "In this scenario, we wish to see whether our system can identify the diagonal movements or not." -> ¿?
Page 14, line 391: "Based on preliminary results, regression, artificial neural networks and KNN were tested." -> According to the previous text, more techniques were tested. Which preliminary results? (This line corresponds to the Methodology section).
Page 15, line 396: "During the testing phase, the subjects are required to wear a smartphone to the lower end of the leg. First the system will auto-calibrate, by using compass data it will assume that the user is facing front. All the future rotations will be calculated based on the initial value of the front. The smartphone communicates with a PC via WIFI and transmits real time IMU data. These data are processed by a trained ANN which outputs a number from 1 to 5 that corresponds to the desired direction. " -> This paragraph corresponds to the Methodology section; nevertheless, it is very repetitive.
Page 15, line 400: "The ANN can estimate the desired direction with 84.1% accuracy, within 0.0192milliseconds. " -> What does this result imply?
Page 15, line 401: "The speed of the calculation is important as any introduction of lags (respond delay) will have a very negative impact 402 to the virtual experience. " -> So it is important or not? This line is contradictory.
Page 15, line 403: "A software application is responsible for simulating the press of the corresponding keyboard keys (such as ‘w’,’s’,’a’,’d’) that will enable the virtual avatar to move in the virtual environment." -> This line corresponds to the Methodology section.}
Page 15, line 406: "however after a short period of time the user gets used to the valid physical locations/poses and tend to place his/her leg at the right position. " -> According with which result?
Page 15, line 410: "Detection of jumping and crouching as well as speed cannot be identified by the current version of the system. Jumping could be implemented by moving the leg in an upward position for the duration of a jump while crouching can be implemented by bending the leg forward. "-> This line is out of context, it should be eliminated.
Page 15, line 413: "The speed could be calculated by moving the leg to the extreme ends of the corresponding movement. For example, moving the leg to a full forward position could mean run forward. The running and walking is important for a lot of games however most of them support only running forward, usually by pressing the “shift” and “w” keys at the same time. Hence this is the most important direction for speed. All these movements can be implemented once additional data are collected, but the current work is significant as it is the first to explore this new way of walking in virtual environments." -> This paragraph does not correspond to the Discussion, it could be included in a Future work section.
Page 15, Section Conclusion: This section needs to be rewritten since the conclusions of this work are not being presented.
Page 15, line 421: "This paper presents a new way for walking in a virtual environment." -> Is this the contribution of this work?
Page 15, line 433: "For example, front-left will result to front and left which is desired as the system can simulate the pressing of two buttons at the same time. Another limitation is that the direction is calculated after the land of the leg to a position. This forced the identification process to start after the user’s physical movement causing 436 a small delay. In order to provide an even..." -> This paragraph does not correspond to this section, it should be eliminated.
Author Response
Reviewer 3 | |
Page 1, line 29: "Oculus Rift and Oculus touch" -> References? | Addressed in the paper, as suggested. |
Page 3, line 114: "which is considered good" -> in comparison with which other error value? | Addressed in the paper, as suggested. |
Page 3, Section Physical Equipment: Which specific mobile equipment was used? | Samsung S8 |
Page 4, line 180: "X = independent variable" -> There is not a X variable in the equation. | X represents various independent variables such as X1, …X9 |
Page 4, line 180: What does β9 mean? | β9 is the slope due to X9 |
Page 4, Section Machine Learning Techniques: Artificial Neural Networks are not described among the techniques presented in this section, which is very important taking into account that is the technique that presented the best results. | Section 3.3.2 added |
Page 5, line 194: "the processing time is slightly longer." -> In comparison with what? | Based on the ref [22], this statement is modified as “But, the processing time to fit for many neighbors can be time consuming” |
Page 5, line 205: Reference? | Reference [22] addresses this. |
Page 6, line 228: "The computational time of the algorithm training is also calculated, but not taken into account..." - > What was the purpose of calculating the computational time? | Computational time of the algorithm is of two types here: for training and testing. The training computational time has no impact on the gameplay as it is the testing time which is valid while the game is being played. |
Page 6, line 236: "The acquired data is pre-processed for normality" -> What this line refers to? Which were the steps for the data preprocessing? | Preprocessing is done for normality check using outlier removal approach. This is mentioned in the same paragraph. |
Page 6, line 237: "outliers which are 1.5 times Inter-Quartile Range away from the first and third Quartile..." -> What happen with those outliers? | Outliers are removed as mentioned in Section 5. |
Page 7, Figure 1: This Figure needs to be better described, specifying in more detail each block of the architecture. | Section 4 Methodology section is elaborated |
Page 7, line 244: "The android mobile application that was developed..." -> How was this application developed? It is necessary to explain the details of the application. | Class information added. |
Page 7, Figure 2: The foot figure does not describe what is presented in the image, besides it needs to be located just below the figure. It is necessary to better describe this figure, specifying each of its parts. | Addressed in the paper, as suggested. |
Page 8, line 268: "The data without any pre-processing gave results as shown in Table 1." -> It is necessary to mention in the Methodology section that the testing of the data is performed using preprocessed and not preprocessed data. On the other hand, what do these results represent? They are results of what? | Section 4 adds “… third Quartile are removed. This preprocessed data is further processed through…” The results in Table 1 are used to show the importance of preprocessing. Else, the accuracy is degraded. |
Page 8, line 272: "As the accuracy did not meet our expectations..." -> Which were those expectations? | Modified in the paper. |
Page 9, line 302: "d=1.45693+0.18771 ×ax+0.19714 ×ay−0.06890 ×??" -> What does it represent? It is not described in the text. | 1.45693 is the intercept and the slope values of various independent variable are the following values. |
Page 9, line 305: It is necessary to mention in the last sections that a non-linear technique is also used, as well as the description of the technique. | Added to Discussion section after Table 4. |
Page 9, line 310: What does this equation represent? | 0.5649771 is the intercept and the slope values of various independent variable are the following values. |
Page 9, line 312: "204 degrees of freedom." -> It is necessary to explain why was chosen this number of degrees of freedom. | Degrees of freedom (df) used in hypothesis testing are related to the observations and the number of independent variables. These are not related to the angular degrees of freedom. |
Page 10, line 331: What does Figure 5 represent? It is not described in the text. The foot figure does not describe the image. | Figure 5 is explained above the figure. |
Page 11, line 333: Which are does classes? How are they represented in the confusion matrix? | Classes represent directions (left, right, …). Added in the paper above Figure 6. |
Page 11, line 337: "Overall training results indicate 84.2% 337 accurate classification..."-> Where is this results indicated? | Figure 6 “Training Confusion Matrix” shows the result. |
Page 11, line 337: "The testing results were also very close with 84.1%, using 15.9% of the dataset of testing."-> Where are those results indicated? | Figure 6 “Testing Confusion Matrix” shows the result. |
Page 13, line 372: "AdaBoost took 2.250457 seconds while Bagging took 2.159735 seconds to train the classifier." -> If the processing time of an algorithm is mentioned, it is necessary to mention the one of all the used ones; however, if the processing time did not fulfill any function for the purpose of this work, it is not necessary to mention it. | Actual values are provided in Tables 2 and 3. As indicated before, computation time is for training and testing. |
Page 14, line 379: "with reasonable computation time." -> Which is that time? Why is it necessary to mention this since the computation time was not taking into account for the selection of an algorithm? | Computation time of the testing phase is vital for algorithm selection. Added in Section in 3.3. |
Page 14, line 391: "Based on preliminary results, regression, artificial neural networks and KNN were tested." -> According to the previous text, more techniques were tested. Which preliminary results? (This line corresponds to the Methodology section). | Based on the results obtained without diagonal movements, regression, … Text changed. |
Page 15, line 400: "The ANN can estimate the desired direction with 84.1% accuracy, within 0.0192milliseconds. " -> What does this result imply? | Implies that ANN can detect the positions accurately. And, it took only 0.0192 milliseconds to generate the output during the testing phase. |
Page 15, line 401: "The speed of the calculation is important as any introduction of lags (respond delay) will have a very negative impact 402 to the virtual experience. " -> So it is important or not? This line is contradictory. | The speed is important, but it is only about the testing computation time. Meanwhile, the training computation time is not important for our work. |
Page 15, line 406: "however after a short period of time the user gets used to the valid physical locations/poses and tend to place his/her leg at the right position. " -> According with which result? | Based on the visual movements in the virtual world. |
Page 15, line 421: "This paper presents a new way for walking in a virtual environment." -> Is this the contribution of this work? | One of the contributions. Other contributions are mentioned in the abstract: I) Synchronizes actions-movements using suitable multiple sensor units, II) Selects the significant features and an appropriate algorithm to process them. |
Typographical Concerns | |
Page 1, line 25: There could be included more keywords, such as "feature selection", "movement identification", among others. | Addressed in the paper, as suggested. |
Page 1, line 29: "virtual reality" -> VR | Addressed in the paper, as suggested. |
Page 1, line 34: "We aim to provide an efficient, low-cost solution that offers a more natural experience to virtual locomotion. Our solution utilizes" -> Before explaining the solution proposed in the presented work, it is necessary to present and develop the problem. This paragraph should be relocated. | We believe that this misconception is solved after reordering certain parts of the introduction, as mentioned by Reviewer 3. |
Page 1, line 43: "This provides a more realistic solution than 43 keyboard/mouse or handheld controllers approaches and yet more resource efficient solution that virtual reality treadmills. " -> Reference? | This is the claim of the authors, as usually people move using their legs and not their hands (keyboard, mouse). |
Page 3, line 138: "Internal Measurement Unit (IMU)" -> It is not necessary to specify a term more than once. | Addressed in the paper, as suggested. |
Page 4, line 175: "to identify the relationship" | Addressed in the paper, as suggested. |
Page 4, line 175: Reference? | Addressed in the paper, as suggested. |
Page 4, line 178: "Linear regression is used to find linear relationship between the dependent and independent variables." -> This line is repetitive. | Addressed in the paper, as suggested. |
Page 4, line 180: "? = ?0 + ?1?1 + ⋯ + ?9?9 + ? " -> It could be helpful to list the equations in order to facilitate their referencing. | Addressed in the paper, as suggested. |
Page 5, line 183, 184, 185, 186, 191: List the equations. | Addressed in the paper, as suggested. |
Page 5, line 184, 185: What do " " and " " mean? | Addressed in the paper, as suggested. |
Page 5, line 193: "KNN" -> K-Nearest Neighbor (KNN) | Addressed in the paper, as suggested. |
Page 5, line 195: "u", "v" -> variables should be written in cursive letter to differentiate them from the rest of the text. | Addressed in the paper, as suggested. |
Page 5, line 199: "u", "v" -> Change to cursive letter | Addressed in the paper, as suggested. |
Page 5, line 200: List the equation | Addressed in the paper, as suggested. |
Page 5, line 203: List the equation | Addressed in the paper, as suggested. |
Page 5, line 208: List the equation. What do "c" and "i" mean? | Addressed in the paper, as suggested. |
Page 5, line 209: List the equation. What do "X" and "T" mean? | Addressed in the paper, as suggested. |
Page 6, line 215: List the equation. | Addressed in the paper, as suggested. |
Page 6, line 218: "AdaBoost algorithm uses an adaptive boosting approach." -> Reference? | Addressed in the paper, as suggested. |
Page 6, line 223: List the equation. What does "T" mean? | Addressed in the paper, as suggested. |
Page 6, line 230: List the equation. | Addressed in the paper, as suggested. |
Page 6, Section Methodology: This section needs to be rewritten, it is very difficult to follow the thread of the text, it is not organized and it does not describe all the steps that were followed, according to the methodology described in the previous sections. | Section 4 Methodology section is elaborated. |
Page 6, line 237: "The data is processed through different machine learning techniques for the purpose of selecting the best fit." -> This line is repetitive. | Addressed in the paper, as suggested. |
Page 6, line 238: "An Artificial Neural Network was selected as the most suitable machine learning technique due to its highest accuracy, 84.1% (on testing data) and 84.2% on training data. " -> This line corresponds to the conclusions section. | Addressed in the paper, as suggested. An Artificial Neural Network was selected as the most suitable machine learning technique due to its highest accuracy, 84.1% (on testing data) and 84.2% on training data. |
Page 8, line 256: "the S key..." | Addressed in the paper, as suggested. |
Page 8, line 259: "virtual reality" -> VR | Addressed in the paper, as suggested. |
Page 8, Section Results: This section needs to be rewritten, it is very difficult to follow the thread of the text, it is not organized and it does not describe the results obtained. It contains a lot of information that corresponds to another sections, such as the methodology, discussion and conclusions sections. | Addressed in the paper, as suggested. |
Page 8, line 265: "Data obtained from IMU indicate different positions of the subject’s leg. This research uses different techniques to classify the given dataset into their respective targeted movements and to ..." -> This text corresponds to the Methodology section. | Moved to methodology section |
Page 8, line 273: "The outlier removal is based on the boxplot approach, which removes any value that is far from the 1st and 3rd quartile by 1.5 times the inter-quartile distance. This pre processing approach ..." -> This paragraph corresponds to the Methodology section. | Moved to methodology section |
Page 8, line 281: "In order to find the relationship between the independent variables (like ax, ay, az, gx, gy, gz, cx 281 and cz) and the dependent variable (d), regression was used. " -> This line is repetitive. | Addressed in the paper, as suggested. |
Page 8, line 288: "The adjusted R2 value" -> It should be "R^2" | Addressed in the paper, as suggested. |
Page 8, line 288: "which is low and thus the applicability of linear regression for this dataset is questionable." -> This line corresponds to the Discussion section. | Results and Discussion sections are combined to make the flow easier |
Page 8, line 289: "To verify this outcome, a normal Q- Q plot was tested." -> This line corresponds to the Methodology section. | Addressed in the paper, as suggested. |
Page 9, line 293: "From the normal Q-Q plot, the normality assumption required for linear regression is not satisfied." -> This line corresponds to the Discussion section. | Results and Discussion sections are combined to make the flow easier |
Page 9, line 289: "To confirm the normality test, a Shapiro-Wilk normality test was performed. The p-value of 294 3.856e-10, which is lower than the alpha value, indicates that the null hypothesis (that the sample comes from a population that is normally distributed) is rejected..." -> This paragraph corresponds to the Methodology section. | Addressed in the paper, as suggested. |
Page 9, line 299: "Further, stepwise regression was used to identify the salient independent variables and see if the adjusted R2 value could..." -> This paragraph corresponds to the Methodology section. (It should be "R^2"). | Addressed in the paper, as suggested. |
Page 9, line 302: "adjusted R2 value" -> It should be "R^2" | Addressed in the paper, as suggested. |
the Methodology section. (It should be "R^2"). | Addressed in the paper, as suggested. |
Page 9, line 306: "As the number of dependent outcomes are five, binomial based Logistic regression could be not applied. Thus, non-linear regression ..." -> This paragraph corresponds to the Methodology section. | Moved to methodology section |
Page 9, line 299: "so the model seems to fit the dataset. The residual plot 313 indicates that the chosen regression is suitable, but there are some outliers which needs to be addressed..." -> This paragraph corresponds to the Discussion section. | Results and Discussion sections are combined to make the flow easier |
Page 10, line 318: "ANOVA was applied on the results of the Poisson regression to understand their applicability." -> This line corresponds to the Methodology section. | Moved to methodology section |
Page 10, line 319: "The obtained Pseudo R2 value" -> It should be "R^2". | Addressed in the paper, as suggested. |
Page 10, line 320: "This indicates a lack of fit for this regression. Thus, we move on to apply other machine learning techniques." -> This line corresponds to the Discussion section. | Results and Discussion sections are combined to make the flow easier |
Page 10, line 323: "Artificial Neural networks with five hidden layers were used since the system has 5 target outcomes. The target position represented in integer form is converted into five binary bits, each bit representing a single integer value..." -> This paragraph corresponds to the Methodology section. | Moved to methodology section |
Page 10, line 327: "During the training process, both the training and the validation datasets were used. During the training phase ..." -> This paragraph corresponds to the Methodology section. | Moved to methodology section |
Page 11, line 341: "K-Nearest neighbors" -> KNN | Addressed in the paper, as suggested. |
Page 11, line 341: "K-Nearest neighbors was applied using Weighted KNN, Cubic KNN and Cosine KNN. By varying the number of neighbors, ..." -> This paragraph corresponds to the Methodology section. | Moved to methodology section |
Page 11, line 347: Table 1 is not cited in the text. | Addressed in the paper, as suggested. It was noted the two tables are labelled as Table 1. |
Page 11, line 349: "Comparing the accuracies, weighted KNN outperformed other approaches. Weighted KNN achieved an overall accuracy of 74.6% when ..." -> This paragraph corresponds to the Discussion section. | Results and Discussion sections are combined to make the flow easier |
Page 12, line 354: Figure 7 and Figure 8 are not mentioned nor described in the text. | Addressed in the paper, as suggested. |
Page 12, line 358: "Three variants of decision trees, such as Simple (with number of splits equal to 4), Medium (with 358 number of splits equal to 20) and Complex (with number of splits equal to 100) Trees were tested." -> This line corresponds to the Methodology section. | Moved to methodology section |
Page 12, line 360: "Medium Trees resulted in better accuracy than other two, with 71.7%, while Simple Tree had an accuracy of 62.4% and that of Complex Trees was 71.4%." -> This line corresponds to the Discussion section. | Results and Discussion sections are combined to make the flow easier |
Page 12, line 362: Figure 9 and Figure 10 are not mentioned nor described in the text. | Addressed in the paper, as suggested. |
Page 13, line 366: "which gave quite similar accuracy to that of the Complex Tree." -> This line corresponds to the Discussion section. | Results and Discussion sections are combined to make the flow easier |
Page 13, line 367: It is difficult to see the text that is inside Figure 11. | Addressed in the paper, as suggested. |
Page 14, line 374: Figure 12 is not mentioned nor described in the text. | Addressed in the paper, as suggested. |
Page 14, Section Discussion: This section needs to be rewritten. The discussion of the results is not presented in an adequate way, besides the section is incomplete. It is important to restructure it, eliminate the text that does not correspond and keep an order that is consistent with the results. | Results and Discussion sections are combined to make the flow easier. Certain parts are also rewritten. |
Page 14, line 378: "Based on the results shown" | Addressed in the paper, as suggested. |
Page 14, line 382 and 384: Table 2 and Table 3 belong to the Results section, besides, these tables do not present the correspondent results of the regression technique. | Results and Discussion sections are combined to make the flow easier. Regression is not provided here as regression is eliminated due to lack of fit. |
Page 14, line 390: "In this scenario, we wish to see whether our system can identify the diagonal movements or not." -> ¿? | Diagonal movement like Front-Right, Front-Left and so on. |
Page 15, line 396: "During the testing phase, the subjects are required to wear a smartphone to the lower end of the leg. First the system will auto-calibrate, by using compass data it will assume that the user is facing front. All the future rotations will be calculated based on the initial value of the front. The smartphone communicates with a PC via WIFI and transmits real time IMU data. These data are processed by a trained ANN which outputs a number from 1 to 5 that corresponds to the desired direction. " -> This paragraph corresponds to the Methodology section; nevertheless, it is very repetitive. | Moved to methodology section. Also, repetition is removed. |
Page 15, line 403: "A software application is responsible for simulating the press of the corresponding keyboard keys (such as ‘w’,’s’,’a’,’d’) that will enable the virtual avatar to move in the virtual environment." -> This line corresponds to the Methodology section.} | Moved to methodology section. Also, repetition is removed. |
Page 15, line 410: "Detection of jumping and crouching as well as speed cannot be identified by the current version of the system. Jumping could be implemented by moving the leg in an upward position for the duration of a jump while crouching can be implemented by bending the leg forward. "-> This line is out of context, it should be eliminated. | A Subsection named “Limitations and Future Work” added. |
Page 15, line 413: "The speed could be calculated by moving the leg to the extreme ends of the corresponding movement. For example, moving the leg to a full forward position could mean run forward. The running and walking is important for a lot of games however most of them support only running forward, usually by pressing the “shift” and “w” keys at the same time. Hence this is the most important direction for speed. All these movements can be implemented once additional data are collected, but the current work is significant as it is the first to explore this new way of walking in virtual environments." -> This paragraph does not correspond to the Discussion, it could be included in a Future work section. | A Subsection named “Limitations and Future Work” added. |
Page 15, Section Conclusion: This section needs to be rewritten since the conclusions of this work are not being presented. | Results and Discussion sections are combined to make the flow easier.
|
Page 15, line 433: "For example, front-left will result to front and left which is desired as the system can simulate the pressing of two buttons at the same time. Another limitation is that the direction is calculated after the land of the leg to a position. This forced the identification process to start after the user’s physical movement causing 436 a small delay. In order to provide an even..." -> This paragraph does not correspond to this section, it should be eliminated. | A Subsection named “Limitations and Future Work” added. |
Round 2
Reviewer 1 Report
Although there have been improvements in the text, the authors still fail to satisfactorily address the following issues:
> The paper also lacks definitions for (apparently important) concepts: what is a pseudo-movement?
The authors state "The main difference between this and other approaches is that our approach captures the pseudo-movement and not the actual movement."
- What does this mean? Why is the proposed approach different from any other gesture-based locomotion?
Additionally, the authors state "Data collected from lower parts of the body, such as the foot or leg, are more sensitive to phases of the walking cycle [15] and are more reliable [16]. The data are then analyzed by machine learning approaches which determines the pseudo movement of the user to move towards a desired direction."
- I find this a bit misleading as it seems to imply that the authors are trying to capture at least some part of the walking cycle. This is not the case.
> It is also not clear why a machine-learning technique is required in this situation. The technique seems to detect only static leg poses, which correspond to different tilt angles of the mobile device.
- Android devices have an API to get the orientation of the device. Why do we need an ML technique here?
> One important aspect of gesture detection for locomotion is the recognition delay: after the user decides to move, how long will it take the system to recognize that intention? (i.e., how long after the users starts moving his leg.)
- Table 3 shows execution time. This is not the same as the time it takes for the user to form the intention to move and the system reacting to that intention. After forming the intention to move the user must position his/her leg in the right pose. This takes time, has it been measured?
I still consider this interesting work, but it seems the focus is on the wrong place. Before stating that "this research points to a new way for solving a major problem in virtual reality.", I think the technique should be evaluated for usability and comfort and compared with other techniques.
Author Response
Reviewer 1 > The paper also lacks definitions for (apparently important) concepts: what is a pseudo-movement? Pseudo-movement is now defined in the introduction. “Within the context of this paper, we define a pseudo-movement as a limited motion of the user’s leg which translates to a 3D vector representing direction, speed and an acceleration of the virtual avatar. The mapping between the pseudo-movement and the movement in the virtual environment does not have to be linear” The authors state "The main difference between this and other approaches is that our approach captures the pseudo-movement and not the actual movement." - What does this mean? Why is the proposed approach different from any other gesture-based locomotion? Addressed in the introduction “The proposed approach falls within the family of gesture-based locomotion, employing innovative leg gestures that require minimal physical movement with minimal hardware requirements.” We provide a more realistic solution than keyboard/mouse or handheld controllers approaches and yet is a more resource efficient solution than virtual reality treadmills. Additionally, the authors state "Data collected from lower parts of the body, such as the foot or leg, are more sensitive to phases of the walking cycle [15] and are more reliable [16]. The data are then analyzed by machine learning approaches which determines the pseudo movement of the user to move towards a desired direction." - I find this a bit misleading as it seems to imply that the authors are trying to capture at least some part of the walking cycle. This is not the case. To make this more clear, we have modified this explanation as follows: “Our solution utilizes the Internal Measurement Unit (IMU) sensors of smartphones placed at the lower parts of user’s legs, for capturing their legs poses. The placement of sensors on lower parts of legs provides more discriminative and reliable data which is easier to classify as seen in [15] and [16], who employed a similar sensor positioning approach in order to capture the phases of the walking cycle. “ > It is also not clear why a machine-learning technique is required in this situation. The technique seems to detect only static leg poses, which correspond to different tilt angles of the mobile device. - Android devices have an API to get the orientation of the device. Why do we need an ML technique here? The following text was included in section 3.3: The initial attempt to solve this problem was in fact implemented based on your observation. However, as different users may move their leg to different positions, a static set of rules based only on mobile device angles proved insufficient. Instead we utilized the API to collect full 9 axis gyroscope data (including compass which was used to calibrate the orientation), which resulted in higher accuracy when used in conjunction with ML. > One important aspect of gesture detection for locomotion is the recognition delay: after the user decides to move, how long will it take the system to recognize that intention? (i.e., how long after the users starts moving his leg.) This information can be found in Table 5, runtime execution. - Table 3 shows execution time. This is not the same as the time it takes for the user to form the intention to move and the system reacting to that intention. After forming the intention to move the user must position his/her leg in the right pose. This takes time, has it been measured? Table 4, shows the training time. Table 5, shows the time that it takes for the system to compute the desired movement. Please note, that we have not measured how long it takes for a user to move his/her leg. I still consider this interesting work, but it seems the focus is on the wrong place. Before stating that "this research points to a new way for solving a major problem in virtual reality.", I think the technique should be evaluated for usability and comfort and compared with other techniques. The results of the user survey are now included in Table 1. The questionnaire was administered during the user data collection phase, but the results were just included. Please see table 1 and its corresponding explanation, which details the results of the questionnaire.
Reviewer 2 Report
My main concern with the paper is still the fact that I do not think that the system offers a more natural experience for walking simulation. Firstly, the user has to be sit down. Secondly, the leg is just used as a four direction stick. Why is this necessary if most of the VR controllers usually include thumbsticks for the exact same purpose? The approach might be viable, but it is not cheaper nor more natural than others available in the literature. I'd recommend the authors to focus on a specific problem their approach makes an improvement.
I still don't know which of the poses are more difficult to be recognized, since I don't know which numeric labels in Figure 7 and 8 correspond to each direction.
Regarding to the representation of the multiclass recognition problem, I still don't see the need of converting five different labels ("front", "back"...) to five binary bits and then converting them again to integer values. I don't think the reference [26] supports this assertion. Do the authors mean that they have employed an ANN with five different binary output nodes? More information about the design of the ANN is needed, since it has been resulted to be the best technique.
Author Response
Reviewer 2 My main concern with the paper is still the fact that I do not think that the system offers a more natural experience for walking simulation. Firstly, the user has to be sit down. Secondly, the leg is just used as a four direction stick. Why is this necessary if most of the VR controllers usually include thumbsticks for the exact same purpose? The approach might be viable, but it is not cheaper nor more natural than others available in the literature. I'd recommend the authors to focus on a specific problem their approach makes an improvement. The results of the user survey are now included in Table 1. The questionnaire was administered during the user data collection phase, but the results were just included in order to address your concerns. The questionnaire was included in the initial submission of the paper. Based on the data it is clear that the proposed approach was a welcomed addition as a gaming controller. We consider our approach to be cheaper as it does not require dedicated hardware since smartphones are commonplace nowadays. I still don't know which of the poses are more difficult to be recognized, since I don't know which numeric labels in Figure 7 and 8 correspond to each direction. The labels have been linked to the directions now in the explanations for figures 7 and 8, in section 5.2. “The classes here represent different directions (front (1), back (2), right(3), middle(4) and left(5)).” Regarding to the representation of the multiclass recognition problem, I still don't see the need of converting five different labels ("front", "back"...) to five binary bits and then converting them again to integer values. I don't think the reference [26] supports this assertion. Do the authors mean that they have employed an ANN with five different binary output nodes? More information about the design of the ANN is needed, since it has been resulted to be the best technique. There is only one ANN, with 5 possible outputs, 1,2,3,4,5 (dependant values), which correspond to the 5 possible directions of movement. The “front”,”back” … are the directions where the virtual avatar will move. For example, if the output is 1, then the character will move front by pressing the ‘w’ key.
Reviewer 3 Report
Navigating virtual environments using leg poses and smartphone sensors
Page 1, line 40: "virtual reality" -> VR
Page 2, line 61: "This experience can be further enhanced by adding walking clues such as audio messages, as proposed in [14], but this is out of the scope of this research." -> This line does not correspond to the Introduction section, it could be parte of the future work.
Page 4, Equation 1: The definition of the variables is not well described. X9 and β9 are not mentioned. A correct form to describe those variables could be, Xi and βi, where i is the number of independent variables.
Page 5, Equation 3: What does β represent?
Page 5, Equation 4: What does i represent?
Page 5, line 202: That equations are not numbered.
Page 7, Equation 11: Why is this equation in bold lettters. What does N represent.
Page 10, Table 1: The label of the tables has to be located in the header.
Page 13, Figures 7 and 8: It is very difficult to distinguish the numbers in the images. Besides, there is not discussion of them.
Page 14, Figures 9 and 10: It is very difficult to distinguish the numbers in the images. Besides, there is not discussion of them.
Page 14, Figure 11: It is very difficult to distinguish the numbers in the images. Besides, there is not discussion of them.
Page 15, Figure 12: It is very difficult to distinguish the numbers in the images. Besides, there is not discussion of them.
Page 15, Tables 3 and 4: The label of the tables has to be located in the header. Table 3 is not discussed in the text.
Page 15, line 418: "Artificial Neural Network" -> ANN
Page 16, line 433: This subsection should be relocated as a section after the Conclusion section.
Author Response
Reviewer 3 Page 1, line 40: "virtual reality" -> VR changed Page 2, line 61: "This experience can be further enhanced by adding walking clues such as audio messages, as proposed in [14], but this is out of the scope of this research." -> This line does not correspond to the Introduction section, it could be parte of the future work. addressed Page 4, Equation 1: The definition of the variables is not well described. X9 and β9 are not mentioned. A correct form to describe those variables could be, Xi and βi, where i is the number of independent variables. Addressed Page 5, Equation 3: What does β represent? Page 5, Equation 4: What does i represent? Page 5, line 202: That equations are not numbered. Page 7, Equation 11: Why is this equation in bold lettters. What does N represent. Page 10, Table 1: The label of the tables has to be located in the header. Addressed Addressed Addressed Fixed Fixed Page 13, Figures 7 and 8: It is very difficult to distinguish the numbers in the images. Besides, there is not discussion of them. Resized. The discussion is before the figures Page 14, Figures 9 and 10: It is very difficult to distinguish the numbers in the images. Besides, there is not discussion of them. Page 14, Figure 11: It is very difficult to distinguish the numbers in the images. Besides, there is not discussion of them. Resized. The discussion is before the figures Page 15, Figure 12: It is very difficult to distinguish the numbers in the images. Besides, there is not discussion of them. Page 15, Tables 3 and 4: The label of the tables has to be located in the header. Table 3 is not discussed in the text. Page 15, line 418: "Artificial Neural Network" -> ANN Resized. The discussion is before the figures The discussion is before the tables Addressed Page 16, line 433: This subsection should be relocated as a section after the Conclusion section. Addressed
Round 3
Reviewer 2 Report
All reported issues have been addressed by the authors.
Reviewer 3 Report
Authors has the majority of the suggestions and changes proposed.